# Peptide Therapeutics: Unveiling the Potential against Cancer—A Journey through 1989

**DOI:** 10.3390/cancers16051032

**Published:** 2024-03-02

**Authors:** Othman Al Musaimi

**Affiliations:** 1School of Pharmacy, Faculty of Medical Sciences, Newcastle upon Tyne NE1 7RU, UK; othman.almusaimi@newcastle.ac.uk or o.al-musaimi@imperial.ac.uk; 2Department of Chemical Engineering, Imperial College London, London SW7 2AZ, UK

**Keywords:** cancer, drugs, antineoplastic, peptides, imaging, theragnostic, tumor, chemotherapy, oncology, ADC, PDC

## Abstract

**Simple Summary:**

The extraordinary growth in the global pharmaceutical industry has extended to include peptides, which are amino acids linked together with an amide bond. Due to their well-tolerable safety profile and specificity, therapeutic peptides offer a means to address unmet medical challenges. A well-known example of a commonly administered peptide is insulin. Peptides are considered excellent complements and, in some cases, preferable alternatives to both small molecules such as paracetamol and very large antibodies. At present, around 100 peptide drugs are available on the global market, with ongoing research yielding over 150 peptides in clinical development and an additional 400–600 peptides undergoing preclinical studies. Peptides play a crucial role in cancer research and treatment, and they can be involved in various aspects of cancer development, detection, and treatment. These medicines demonstrate exceptional efficacy in combating cancer, contributing to improved survival rates for cancer patients.

**Abstract:**

The United States Food and Drug Administration (FDA) has approved a plethora of peptide-based drugs as effective drugs in cancer therapy. Peptides possess high specificity, permeability, target engagement, and a tolerable safety profile. They exhibit selective binding with cell surface receptors and proteins, functioning as agonists or antagonists. They also serve as imaging agents for diagnostic applications or can serve a dual-purpose as both diagnostic and therapeutic (theragnostic) agents. Therefore, they have been exploited in various forms, including linkers, peptide conjugates, and payloads. In this review, the FDA-approved prostate-specific membrane antigen (PSMA) peptide antagonists, peptide receptor radionuclide therapy (PRRT), somatostatin analogs, antibody–drug conjugates (ADCs), gonadotropin-releasing hormone (GnRH) analogs, and other peptide-based anticancer drugs are analyzed in terms of their chemical structures and properties, therapeutic targets and mechanisms of action, development journey, administration routes, and side effects.

## 1. Introduction

Peptides are medium-sized molecules with a molecular weight range of 500 to 5000 Da [1]. Peptides have been always important for the living organism [2,3]. They are involved in several biological processes, including the transduction of other molecules, intercellular communication, among others [4]. In 1882, Curtius carried out the first ever peptide synthesis and synthesized benzoylglycylglycine [5]. In 1901, Fisher synthesized the first free dipeptide and coined the term PEPTIDE [6]. Since then, peptides have consolidated their presence across various fields. Their importance as therapeutic agents gained momentum with the isolation of insulin by Frederick Banting in 1921 [7,8]. Peptides have diversified beyond their natural endogenous human peptides to encompass various sources and have been incorporated into medicinal chemistry to introduce novel modified peptide structures [9,10].

Peptides are competing heavily in the pharmaceutical arena, with global sales of more than USD 70 billion in 2019 [11]. A 10% increase in their compound annual growth rate (CAGR) by 2028 is also foreseen (Figure 1) [11].

Figure 1 demonstrates the intense competition of peptides in the pharmaceutical arena across various applications. These developments have been underpinned by continuous developments and advancements in their synthetic method. The advent of solid-phase peptide synthesis (SPPS) has made peptide synthesis easier and faster and played a significant role in increasing the number of peptides reaching the market [12]. During the last three years, 17 peptides have received FDA approval [13]; this accounts for almost half of the number of approvals during the previous decade. Indeed, this implies that peptides are expected to capture a substantial percentage by 2030, which will probably exceed the border of 60 approvals.

Peptides play a crucial role as therapeutic agents in various fields, including in diagnosis [14], immunology [15], and drug discovery [16]. They are able to act as hormones and neurotransmitters, bind to cell surfaces, and trigger intracellular signals with incredible affinity and specificity, with along low immunogenicity [17,18,19]. They are able to fulfil the requirement for interacting/inhibiting large surfaces such as protein–protein interactions (PPIs) [20,21]. Peptides offered a more precise targeted therapy and significantly improved the survival rate of cancer patients in [22]. They can exert their biological effects through various mechanisms; while some of them can invade cells and exert their actions in the cytoplasm (direct translocation) [23,24], others bind to the cell surface receptors and act through agonism/antagonism mechanisms [25]. The first peptide-based anticancer drug to be approved by the FDA was goserelin (Zoladex) in 1989 [26].

Peptide sequences can be derived from (i) natural sources, (ii) produced by chemical synthesis, and (iii) through a peptide library screening approach [27], where phage display is an effective tool for this purpose [28]. Three main technologies considered for synthesizing peptides are (i) classical solution peptide synthesis (CSPS), (ii) SPPS, and (iii) liquid-phase peptide synthesis (LPPS) [29].

Through an analysis of FDA approvals for anticancer peptide-based drugs, it becomes evident that the initial three decades primarily emphasized the development of GnRH and somatostatin analogs. However, a significant shift occurred in 2016, marking a revolutionary phase with the approval of new classes of peptide-based anticancer drugs. This includes peptide receptor radionuclide therapy (PRRT) in 2016, antibody–drug conjugates (ADCs) in 2019, and prostate-specific membrane antigen (PSMA) in 2020 (Figure 2). This evolution represents a noteworthy diversification in the landscape of peptide-based anticancer therapeutics.

Approximately 29 peptide-based drugs for cancer therapy have received approval from the United States Food and Drug Administration (FDA) (Figure 3). This review will shed light on their chemical structures and properties, development journeys, therapeutic targets, mechanisms of action, and FDA approval and conclude with remarks and personal perspectives regarding these peptides.

## 2. Prostate-Specific Membrane Antigen (PSMA) Peptide Antagonists

PSMA, alternatively referred to as glutamate carboxy-peptidase II (GCPII), is a type II transmembrane zinc metalloenzyme that is highly expressed in the majority of prostate cancers and the neo-vasculature of many solid tumors [30]. Notably, it is not upregulated in the vasculature of normal tissues, making it a valuable biomarker and targeting receptor for various applications [30]. PSMA comprises an intracellular domain from 1 to 18 amino acids, a transmembrane domain from 19 to 43 amino acids, and an extensive extracellular domain from 44 to 750 amino acids [30]. Peptides represent an alternative approach to antibody-based anti-PSMA therapies, where they can selectively bind to PSMA-producing cells and deliver molecules of interest either for imaging and/or treatment purposes [31].

The FDA has approved four PSMA-binding peptides, and one of them is able to exert a therapeutic effect, while the rest are only to be used for imaging purposes with positron emission tomography (PET) (Table 1).

### 2.1. ^68^Ga-PSMA-11 (^68^Ga Gozetotide)

^68^Ga-PSMA-11 comprises acyclic chelator HBED-CC (*N*,*N*′-bis [2-hydroxy-5-(carboxyethyl)benzyl] ethylenediamine *N*,*N′*-diacetic acid), a radionuclide ^68^Ga, and an urea-based peptidomimetic HO-Glu-NH-CO-NH-Lys-OH (Figure 4). This diagnostic peptide is employed for the PET imaging of PSMA-positive lesions in men diagnosed with prostate cancer. It is utilized in cases where there is a suspicion of metastasis in individuals eligible for initial definitive therapy or those showing potential recurrence indicated by increased serum prostate-specific antigen (PSA) levels [32].

^68^Ga-PSMA-11 binds to PSMA, which is excessively expressed by malignant prostate cancer cells via clathrin-coated pits. Then, the complex is internalized, and ^68^Ga emits β+, which allows for PET [32].

The University of California developed ^68^Ga-PSMA-11, and it received FDA approval in 2020 [33]. ^68^Ga-PSMA-11 is administered intravenously and may result in adverse effects such as nausea, diarrhea, and dizziness [32].

### 2.2. Piflufolastat F 18 (Pylarify)

Piflufolastat F 18 is a peptidomimetic agent that comprises a PSMA-11 inhibitor (HO-Glu-NH-CO-NH-Lys-OH) and 6-[^18^F]fluoro-pyridine-3-carbonyl (Figure 5). It is used for the PET imaging of PSMA-positive lesions in men diagnosed with prostate cancer [34].

Piflufolastat F 18 selectively binds to the cells expressing PSMA, including malignant prostate cancer cells that exhibit an overexpression of PSMA. The emitted β+ radiation from the ^18^F radionuclide aids in facilitating PET imaging [34].

Progenics Pharmaceuticals developed Pylarify, and it obtained FDA approval in 2021 [35]. Pylarify is administered intravenously and may cause various side effects, including headache, dysgeusia, and fatigue [34].

### 2.3. Lutetium ^177^Lu Vipivotide Tetraxetan (Pluvicto)

Pluvicto is a radio ligand therapeutic agent that comprises PSMA, HO-Glu-NHCO-NH-Lys-OH, and DOTA (1,4,7,10-tetraazacyclododecane-1,4,7,10-tetraacetic acid) chelator labeled with ^177^Lu (Figure 6). It is administered for the therapy of adult patients diagnosed with PSMA-positive metastatic castration-resistant prostate cancer (mCRPC) who have undergone the inhibition of the androgen receptor (AR) pathway and received chemotherapy based on taxanes [36].

Pluvicto binds to cells that express PSMA, ^177^Lu emits β-emission and delivers it to PSMA cells and the neighboring ones, inducing DNA damage and, finally, cell death [36].

The initial development of pluvicto was a collaborative effort between DKFZ (German Cancer Research Center) and the University Hospital Heidelberg in Heidelberg, Germany [37]. Pluvicto was initially licensed to Advanced Biochemical Compounds, and later, its licensing was transferred to Endocyte [38]. Pluvicto received FDA approval in 2022 [39]. Pluvicto is administered intravenously and has exhibited various side effects, including reduced lymphocytes, decreased hemoglobin, lowered leukocytes, decreased platelets, as well as decreased levels of calcium and sodium [36].

### 2.4. Flotufolastat F 18 (Posluma)

Flotufolastat F 18 is a radioactive diagnostic agent that comprises a PSMA-binding ligand that is a DOTAGA complex with nonradioactive Ga^3+^ and a radioactive ^18^F that is covalently bound to silicon (Figure 7) [40]. It is used in conjunction with the PET imaging of PSMA-positive lesions in men diagnosed with prostate cancer [40]. 

Flotufolastat F 18 binds to PSMA, which is overexpressed in prostate cancer cells. Subsequently, the complex is internalized, and the emitted β+ radiation from the ^18^F facilitates detection through PET imaging [40]. 

Flotufolastat F 18 is the forth PSMA-containing drug to receive approval for the same indication, following Pluvicto [41], Pylarify [42], and ^68^Ga gozetotide [43], which were approved in 2022, 2021, and 2020, respectively. Blue Earth Diagnostics developed Posluma, and it obtained FDA approval in 2023 [44]. Posluma is administered intravenously and has demonstrated adverse effects including diarrhea, increased blood pressure, and injection-site pain [40].

## 3. Peptide Receptor Radionuclide Therapy (PRRT)

PRRT is a successful tool for inoperable or metastasized neuroendocrine tumors (NETs). It comprises a cyclic octapeptide, either octreotide or an octreotate somatostatin peptide analog, radiolabeled with a radionuclide and a chelator [45]. Octreotate comprises Thr at its terminal, whereas it is reduced in the case of octreotide to Thr alcohol. Somatostatin preferentially binds to somatostatin receptor (SSTR) subtypes 2 and 5 [46] with various affinity profiles [46]. DTPA (diethylenetriaminepenta-aceticacid) and DOTA (1,4,7,10-tetraazacyclododecane-1,4,7,20tetra-aceticacid) are commonly used as chelators for PRRT [45]. The crucial characteristics of the radionuclide for targeted therapy include its classification, half-life, energy, particulate radiation branching ratio, photon abundance, and emitted radiation type [47]. The emitted light’s characteristics are pivotal in determining its penetration depth into tumor tissue, thereby influencing its suitability for PRRT [45]. Modifying any of the mentioned components leads to a change in the overall pharmacokinetic and pharmacodynamic profile of the compound. The FDA has granted approval to five PRRTs, one of which is theragnostic, while the remaining four serve diagnostic purposes (Table 2).

### 3.1. Depreotide (Neotect)

Depreotide is a cyclic peptide amide consisting of 10 amino acid residues, and it functions as a somatostatin analog (Figure 8). It is used with Technetium (^99m^Tc) as a diagnostic radiopharmaceutical for conducting scintigraphic lung tumor imaging [48].

Depreotide binds to SSTR expressed in tumor cells, predominantly subtypes 2, 3, and 5. The resulting complex is internalized, and the emission of γ-2 particles by ^99m^Tc facilitates imaging [48,49].

Diatide developed Neotect, which received FDA approval in 1999 [50]. Neotect is administered intravenously and may be associated with side effects such as rash, angioedema, fever, anaphylaxis due to hypersensitivity reactions, a transient increase in blood pressure, seizures, arrhythmias, and syncope. When used in abdominal imaging, abdominal pain, vomiting, and diarrhea may occur [51].

### 3.2. ^68^Ga-DOTATATE(Netspot)

^68^Ga-DOTATATE is a somatostatin analog within the realm of PRRT. It consists of DOTA as the chelating ligand, ^68^Ga as the radionuclide, and the peptide [Tyr^3^]-octreotate (Figure 9). It is used as a diagnostic agent in conjunction with PET to localize SSTR-positive NETs in both adult and pediatric patients [52].

^68^Ga-DOTATATE preferentially binds to SSTR2, which is commonly overexpressed in malignant NET cells. The emitted β+ particles from ^68^Ga enable PET imaging [52,53]. Interestingly, ^68^Ga-DOTATTATE has demonstrated a 10-fold higher in vitro affinity for SSTR2 compared to its counterpart, ^68^Ga-DOTATOC [54].

Advanced Accelerator Applications developed ^68^Ga-DOTATATE, and it received FDA approval in 2016 [55]. ^68^Ga-DOTATATE is administered intravenously and has been associated with adverse effects including nausea, pruritus, and flushing [52].

### 3.3. [^177^Lutetium]Lu-DOTA-TATE (Lutathera)

[^177^Lu]Lu-DOTA-TATE, also known as [[^177^Lu]Lu-DOTA^0^, Tyr^3^]-octreotate, is a somatostatin analog that includes a combination of 1,4,7,10-tetraazacyclododecane-1,4,7,10-tetraacetic acid (DOTA) as the chelating ligand, ^177^Lu as the radionuclide, and the [Tyr^3^]-octreotate peptide (Figure 10). [^177^Lu]Lu-DOTA-TATE is used in the treatment of SSTR-positive gastroenteropancreatic NETs (GEP-NETs), including foregut, midgut, and hindgut NETs in adults [56].

Lutathera preferentially binds to SSTR2, which is overexpressed in tumor cells and subsequently undergoes internalization. The emitted β-emission from the ^177^Lu radionuclide induces cell damage by forming free radicals in the vicinity of SSTR2 and the adjacent cells [47,56,57,58].

Advanced Accelerator Applications developed Lutathera, and it obtained FDA approval in 2018 [59]. Lutathera holds the distinction of being the first FDA-approved PRRT [60]. Lutathera is administered intravenously and has been associated with several adverse effects, including lymphopenia, increased GGT, vomiting, nausea, elevated AST, heightened ALT, hyperglycemia, and hypokalemia [56,61].

### 3.4. ^68^Ga-DOTATOC

^68^Ga-DOTATOC is a somatostatin analog used in PRRT. It is composed of DOTA as the chelating ligand, ^68^Ga as the radionuclide, and the [Tyr^3^]-octreotide peptide (Figure 11). It is used as a diagnostic agent with the PET to localize SSTR-positive NETs in both adult and pediatric patients [62].

^68^Ga-DOTATOC preferentially binds to overexpressed SSTR2 in malignant NET cells. The emitted β+ radiation from ^68^Ga enables emission tomography, facilitating PET imaging [62]. Interestingly, ^68^Ga-DOTATOC has demonstrated the ability to detect more NET lesions with high reproducibility than its counterpart, ^68^Ga-DOTATATE [54].

The University of Iowa Health Care (UIHC) developed ^68^Ga-DOTATOC, and it obtained FDA approval in 2019 [63]. ^68^Ga-DOTATOC is administered intravenously and has exhibited various adverse effects, including nausea, pruritus, and flushing [62].

### 3.5. ^64^Cu-DOTATATE (Detectnet)

^64^Cu -DOTATATE is a somatostatin analog used in PRRT. It consists of DOTA as the chelating ligand, ^64^Cu as the radionuclide, and the [Tyr^3^]-octreotate peptide (Figure 12). It is used as a diagnostic agent with PET to localize SSTR-positive NETs in adult patients [64,65].

Detectnet binds to the SSTRs of cells that overexpress SSTR2. ^64^Cu emits β+ emission and allows for PET imaging [64]. Due to the longer half-life of ^64^Cu compared to ^68^Ga, ^64^Cu-DOTATATE surpasses ^68^Ga-DOTATOC in performance, leading to enhanced lesion detection [66].

Radiomedix developed Detectnet, and it obtained the FDA approval in 2020 [67]. Detectnet is administered intravenously and may cause several side effects, including nausea, vomiting, and flushing [64].

## 4. Somatostatin Analogs

SSTRs play a crucial role in inhibiting the secretion of important hormones that contribute to tumor growth, such as growth hormone and thyroid-stimulating hormone [68]. There are five subtypes of SSTRs, labeled SSTRs 1–5, and they are activated by both somatostatin-14 and somatostatin-28 [69,70]. Their presence in almost all meningiomas positions them as valuable tools in radiodiagnostic, prognostic, and therapeutic applications. Inhibiting these receptors with the assistance of somatostatin analogs proves beneficial in treating recurrent meningiomas. The FDA has approved seven analogs based on somatostatin peptides, and a discussion of five of them was provided earlier in Section 3.

### 4.1. Octreotide (Sandostatin)

Octreotide is an 8-mer cyclic with the sequence H-fCFwKTCT(OH), linked through a disulfide bridge formed between Cys2 and Cys7 (Figure 13). It is used to treat acromegaly and alleviate symptoms associated with metastatic carcinoid tumors (flushing and diarrhea) and vasoactive intestinal peptide (VIP)-secreting adenomas (watery diarrhea) [71].

Octreotide exhibits the same effect as its natural analog, somatostatin. It acts as a potent inhibitor of growth hormone, glucagon, and insulin. Additionally, it inhibits the release of LH in response to GnRH [72]. Octreotide also decreases splanchnic blood flow and inhibits the release of serotonin, gastrin, vasoactive intestinal peptide, secretin, motilin, and pancreatic polypeptide. This comprehensive action provides relief for the gastrointestinal and flushing symptoms associated with carcinoid and/or VIPoma tumors [71,73].

Novartis Pharmaceuticals developed Sandostatin, and it received FDA approval in 1998 [71]. Sandostatin is administered subcutaneously and may be associated with several side effects, including pain at the injection site, fatigue, dizziness, headache, conduction abnormalities, dose-related diarrhea, hypo/hyperglycemia, and hypothyroidism. In 2020, Camargo Pharmaceutical Services introduced Mycapssa, a delayed-release oral octreotide which received FDA approval during the same year [74,75].

### 4.2. Lanreotide (Somatuline)

Lanreotide is an 8-mer cyclic peptide amide functioning as a somatostatin analog. It features a disulfide bridge formed between Cys2 and Cys7 (Figure 14). It is used to treat acromegalic patients who have not responded adequately to, or cannot undergo, surgery and/or radiotherapy. Additionally, it is used to treat adult patients with unresectable, well- or moderately differentiated, locally advanced, or metastatic GEP-NETs with the aim of improving progression-free survival [76].

Lanreotide binds and activates SSTR2 and SSTR5, leading to the inhibition of the production of substances that support tumor growth and inducing cell arrest. Additionally, it inhibits the release of growth hormone in the brain [76]. 

Beafour Ipsen developed Somatuline, and it received FDA approval in 2007 [77]. Somatuline is administered subcutaneously and may be associated with side effects including diarrhea, cholelithiasis, abdominal pain, nausea, injection-site reactions, musculoskeletal pain, vomiting, headache, hyperglycemia, hypertension, and cholelithiasis [76].

## 5. Antibody Drug Conjugate (ADCs)

ADCs are a novel class for cancer treatments that aid the delivery of highly potent cytotoxic anticancer drugs to be released at the therapeutic target [78]. They are composed of three main components, a monoclonal antibody (mAb), linker, and the payload [78]. 

Monoclonal antibodies have been proven to play a central role in cancer therapy. They are produced either by recombinant DNA technology in a mammalian cell line (Chinese Hamster Ovary) or murine hybridoma technology. The selection of the linker to anchor the payload to the mAb is of utmost importance. It is imperative for the linker to be stable in the blood stream and be able to undergo a programmed metabolism to release the payload at the therapeutic target [79,80]. A valine–citrulline (Val-Cit)-cleavable linker is an optimal choice for this purpose. In contrast to other tetrapeptidyl linkers like GFLF and ALAL, Val-Cit has not exhibited any aggregation problems during the conjugation process [79,80,81]. It is worth highlighting that other known linkers do not fulfil the aforementioned prerequisite for an ideal linker. For example, hydrazone-based linkers have demonstrated premature payload release, attributed to instability at different pH values, while disulfide-based linkers have exhibited susceptibility to exchanging with other thiols. Two maleimide-based non-cleavable linkers, namely (i) maleimidocaproyl and (ii) maleimidomethyl cyclohexane1-carboxylate, are commonly utilized. However, despite their widespread use, this linker is susceptible to a retro-Michael reaction, leading to the premature detachment of the drug linker from the ADC [82]. Furthermore, the succinimidyl thioether moiety undergoes irreversible hydrolysis, a reaction that frequently takes place in ADCs, often at rates that surpass their intended in vivo lifetime [83,84,85]. Various strategies are employed to address these issues. One approach involves leveraging the stability of the hydrolyzed form of the succinimidyl thioether. This includes intentionally opening the ring of the succinimidyl thioether before its exposure to the exogenous thiol [86,87]. Thioether-based linkers exhibited delayed release, potentially resulting in the complete loss of payload activity [80,88]. 

Although the amide bond between the linker and the payload maintains stability during blood circulation, selective cleavage can occur through the action of lysosomal cathepsin B or cathepsin L, depending on the nature of the linker (Figure 15).

Two peptides were primarily considered as cytotoxic payloads in the approved ADCs: synthetic monomethyl auristatin E (MMAE) and monomethyl auristatin F (MMAF). Both are potent microtubule-disrupting agents and are structural analogs of the natural dolastatin 10 (Figure 16) [89]. Dolastatin 10 is a potent antineoplastic pentapeptide that was first isolated from the marine *mollusk Dolabella auricularia* by Pettit et al. in 1987 [90]. Dolastatin 10 has the following amino acid residues: Dolavaline (Dov), Val, Dolaisoleuine (Dil), Dolaproine (Dap), and Dolaphenine (Doe) [90,91].

The FDA has approved a total of 11 ADCs and 164 are undergoing clinical studies [92]. Peptides can be found in ADCs either as the payload, the linker, or, in some cases, they may play both roles concurrently [93]. Among the 11 FDA-approved ADCs, 6 incorporate peptides within their structures, serving as either the payload, linker, or both (Table 3) [42,43,93].

### 5.1. Enfortumab Vedotin-Ejfv (Padcev)

Enfortumab Vedotin-Ejfv is an ADC that helps in delivering a cytotoxic chemotherapeutic agent to the tumor [94]. It is composed of three main components: (i) the drug or payload, which is a synthetic agent named monomethyl auristatin E (MMAE) [89,95,96]; (ii) maleimide–hexanoyl (maleimidocaproyl); (iii) a cathepsin-cleavable linker which is a Val-Cit dipeptide linker; and (iv) a mAb that targets Nectin-4. A maleimidocaproyl moiety is introduced at the *N*-terminal of the dipeptide Val-Cit to facilitate conjugation to the antibody (Figure 17) [78]. It is used to treat adult patients with locally advanced or metastatic urothelial cancer who have previously undergone treatment with a programmed death receptor-1 (PD-1) or programmed death-ligand 1 (PD-L1) inhibitor, as well as a platinum-containing chemotherapy in neo-adjuvant/adjuvant, locally advanced, or metastatic settings [97].

The antibody in Padcev is a human IgG1 that specifically targets the Nectin-4 receptor, a cell surface protein crucial for adhesion. This receptor is overexpressed in tumor cells [98]. Following the binding of the ADC to Nectin-4, internalization occurs to form the ADC-Nectin-4 complex. Subsequently, the Val-Cit linker undergoes cleavage, leading to the release of MMAE [81]. MMAE disrupts the microtubule network within the tumor cell, leading to cell arrest and the induction of apoptosis [97].

Astellas Pharma developed Padcev, and it received FDA approval in 2019 [99]. Padcev is administered intravenously and has been associated with various adverse effects, including fatigue, peripheral neuropathy, a decreased appetite, rash, alopecia, nausea, dysgeusia, diarrhea, dry eye, pruritus, and dry skin [97].

### 5.2. Polatuzumab Vedotin-Piiq (Polivy)

Polivy is an ADC drug that shares the same components as Padcev [100]. However, it features an antibody that targets CD79b, which is overexpressed in mature B-cells (Figure 18). It is utilized to treat adult patients with previously untreated diffuse large B-cell lymphoma (DLBCL), not otherwise specified (NOS), or high-grade B-cell lymphoma (HGBL) who have an international prognostic index score of 2 or greater. Polivy is administered in combination with a rituximab product, cyclophosphamide, doxorubicin, and prednisone (R-CHP). Additionally, it is employed in the treatment of adult patients with relapsed or refractory DLBCL, NOS, after undergoing at least two prior therapies. In this case, it is used in combination with bendamustine and a rituximab product [101]. A synergistic effect is anticipated due to the additional impact of the mentioned drugs, each operating through distinct mechanisms. These mechanisms include targeting the CD20 protein on the surface of cancerous cells (rituximab) [102], alkylation and DNA strand breakage (cyclophosphamide) [103], inhibiting topoisomerase II and causing DNA double-strand breakage (doxorubicin) [104], and addressing or preventing conditions associated with cancer, such as anemia, drug hypersensitivity, or hypercalcemia prednisone (R-CHP) [105]. This combination exhibited a 27% reduction in the relative risk of disease progression [105].

The antibody component of the drug binds to CD79b, a B-cell-specific surface protein that is part of the B-cell receptor [106,107]. Upon binding, the complex is internalized. Subsequently, the Val-Cit linker is cleaved [81], leading to the release of MMAE. MMAE binds to microtubules and eliminates the dividing cells through inhibiting cell division and inducing apoptosis [101].

Roche developed Polivy, which obtained FDA approval in 2019 [108]. Polivy is administered intravenously and may cause adverse effects such as peripheral neuropathy, nausea, fatigue, diarrhea, constipation, alopecia, and mucositis [101].

### 5.3. Fam-Trastuzumab Deruxtecan-Nxki (Enhertu)

Fam-trastuzumab deruxtecan-nxki is an ADC that incorporates a peptide solely as a linker. The GGFG tetrapeptide linker is designed to be cleavable by lysozymes (Figure 19) [109]. This ADC combines a human epidermal growth factor receptor-2 (HER2)-directed antibody with a topoisomerase inhibitor conjugate. It is used to treat adult patients with unresectable or metastatic HER2-positive breast cancer, unresectable or metastatic NSCLC with activating HER2 (ERBB2) mutations, and those with locally advanced or metastatic HER2-positive gastric or gastroesophageal junction adenocarcinoma who have previously undergone a trastuzumab-based regimen [110].

Upon binding to HER2 on malignant cells, trastuzumab deruxtecan undergoes internalization. Subsequently, linker cleavage is initiated by lysosomal enzymes. Upon release through cleavage, trastuzumab deruxtecan induces targeted DNA damage and apoptosis in cancer cells, facilitated by its ability to cross cell membranes [110].

Daiichi Sankyo developed Enhertu, which obtained FDA approval in 2019 [111]. Enhertu is administered intravenously and may lead to adverse effects including nausea, a decreased white blood cell count, decreased hemoglobin, a decreased neutrophil count, a decreased lymphocyte count, fatigue, a decreased platelet count, increased aspartate aminotransferase, vomiting, increased alanine aminotransferase, alopecia, increased blood alkaline phosphatase, constipation, musculoskeletal pain, a decreased appetite, hypokalemia, diarrhea, and respiratory infection. For gastric cancer, adverse effects may include decreased hemoglobin, a decreased white blood cell count, a decreased neutrophil count, a decreased lymphocyte count, a decreased platelet count, nausea, a decreased appetite, increased aspartate aminotransferase, fatigue, increased blood alkaline phosphatase, increased alanine aminotransferase, diarrhea, hypokalemia, vomiting, constipation, increased blood bilirubin, pyrexia, and alopecia [110].

### 5.4. Belantamab Mafodotin-Blmf (Blenrep)

Belantamab Mafodotin-Blmf is an ADC drug that comprises afucosylated humanized immunoglobulin G1 mAb (IgG1), a protease-resistant maleimidohexanoic linker, and monomethyl auristatin F (MMAF) as the payload (Figure 20) [89]. It is used in the treatment of adult patients with relapsed or refractory multiple myeloma (a blood cancer that develops in plasma cells in the bone marrow) who have undergone at least four prior therapies including an anti-CD38 mAb, a proteasome inhibitor, and an immunomodulatory agent [112].

Afucosylated IgG1 binds to B-cell maturation antigen (BCMA), which is expressed in both normal B lymphocytes and multiple myeloma cells. Subsequently, the complex is then internalized. As the linker is non-cleavable, MMAF is released through the proteolysis of the mAb component. MMAF binds and disrupts the microtubule network, causing cell cycle arrest and apoptosis [112,113].

GlaxoSmithKline developed Blenrep, which obtained FDA approval in 2020 [114]. Blenrep is administered intravenously and has shown various side effects, including (≥20%) keratopathy (corneal epithelium change on eye exam), decreased visual acuity, nausea, blurred vision, infusion-related reactions, pyrexia, and fatigue. Side effects occurring in at least 5% of cases include a decrease in platelet, neutrophil, and lymphocyte counts; decreased hemoglobin; decreased gamma-glutamyl transferase; and increased creatinine [112].

### 5.5. Tisotumab Vedotin-Tftv (Tivdak)

Tisotumab Vedotin-Tftv is an ADC drug that includes an antibody-targeting tissue factor (TF-011), also recognized as thromboplastin, factor III, or CD142. It is combined with a Val-Cit linker and a payload of MMAE (Figure 21). It is used in the treatment of adult patients with recurrent or metastatic cervical cancer experiencing disease progression upon or after chemotherapy [115].

Tisotumab Vedotin-Tftv binds to the tissue factor TF011, forming a complex that is subsequently internalized. The Val-Cit linker undergoes cleavage, leading to the release of MMAE. MMAE interferes with the microtubule network in dividing cells, causing cell arrest and ultimately inducing apoptosis [115]. Additional mechanisms have been elucidated, including Fc receptor-mediated effector functions such as antibody-dependent cellular toxicity (ADCC) and antibody-dependent cellular phagocytosis (ADCP). Moreover, there is the inhibition of protease-activated receptor-2 (PAR-2)-dependent signaling through the antigen-binding fragment [116].

Tivdak was co-developed by Seagen and Genmab, and it received FDA approval in 2021 [117]. Tivdak is administered intravenously and has demonstrated several adverse effects, including decreased hemoglobin, fatigue, decreased lymphocytes, nausea, peripheral neuropathy, alopecia, epistaxis, conjunctival adverse reactions, hemorrhage, decreased leukocytes, increased creatinine, dry eye, an increased prothrombin international normalized ratio, prolonged activated partial thromboplastin time, diarrhea, and rash [115].

### 5.6. Loncastuximab Tesirine-Lpyl (Zynlonta)

Loncastuximab Tesirine-Lpyl is an ADC drug that includes a humanized IgG1 kappa mAb. It utilizes the Val-Ala-cleavable linker as its sole peptide component within its structure. The mAb is conjugated to SG3199, a pyrrolobenzodiazepine (PBD) dimer cytotoxin, through the Val-Ala linker (Figure 22) [118]. It is used in the treatment of adult patients with relapsed or refractory large B-cell lymphoma after two or more lines of systemic therapy. This includes DLBCL, NOS, DLBCL arising from low-grade lymphoma and high-grade B-cell lymphoma [119].

Upon the binding of IgG1 kappa, mAb binds to CD19, and loncastuximab tesirine undergoes internalization into the cell. Subsequent proteolytic cleavage releases the SG3199 component. SG3199 then binds to the DNA minor groove, forming cytotoxic DNA interstrand crosslinks, ultimately leading to B-cell death [119].

ADC Therapeutics developed Zynlonta, which received FDA approval in 2021 [120]. Zynlonta is administered intravenously and has demonstrated several adverse effects, including thrombocytopenia, increased gamma-glutamyltransferase, neutropenia, anemia, hyperglycemia, transaminase elevation, fatigue, hypoalbuminemia, rash, edema, nausea, and musculoskeletal pain [119]. 

## 6. Gonadotropin-Releasing Hormone (GnRH) Analogs

Gonadotropin-releasing hormone (GnRH) triggers the pituitary secretion of luteinising hormone (LH) and follicle-stimulating hormone (FSH) [121,122]. As a result, GnRH regulates the hormonal and reproductive functions of the gonads [121]. The overexpression of GnRH has been detected in various human cancerous cells, including both reproductive and nonreproductive organs [123]. Manipulating the activation or suppression of this functionality is crucial for fulfilling various requirements. For instance, it is essential in preventing luteinization during assisted reproduction or for treating disorders dependent on sex hormones (Figure 23). 

Although numerous non-peptide GnRH antagonists are available [125], peptides have also established their significance in this important field [126]. They are used to treat prostate cancer by reducing testosterone levels, which are responsible for stimulating the growth of various forms of prostate cancer. This reduction in testosterone contributes to the shrinkage of prostate cancer. Notably, their mechanism avoids hypoestrogenic side effects, flare-ups, or the extended down-regulation period associated with agonists [127]. They preferentially bind to pituitary GnRH receptors and can either enhance or inhibit their release within hours. The FDA has granted approval to nine peptide-based drugs targeting GnRH (Table 4).

### 6.1. GnRH Agonists

#### 6.1.1. Goserelin (Zoladex)

Goserelin is a 10-mer peptide and GnRH agonist (Figure 24). It is used in conjunction with flutamide for the management of locally confined carcinoma of the prostate, as well as for the palliative treatment of advanced carcinoma of the prostate. The utilization of both goserelin and flutamide in combination resulted in an increased rate of response and a faster normalization of elevated levels of prostatic acid phosphatase and prostatic-specific antigen compared to the use of goserelin alone in [128]. Additionally, it is utilized in the management of endometriosis, as an endometrial-thinning agent before endometrial ablation for dysfunctional uterine bleeding, and in the palliative treatment of advanced breast cancer in pre- and perimenopausal women [129].

Goserelin functions by inhibiting pituitary gonadotropin secretion, leading to an initial rise in serum LH and FSH levels. Subsequently, there are subsequent increases in serum testosterone levels [129].

AstraZeneca developed Zoladex, which received FDA approval in 1989 [26]. Zoladex is administered subcutaneously, and it may lead to adverse effects including hot flashes, sexual dysfunction, decreased erections, and lower urinary tract symptoms [129].

#### 6.1.2. Leuprolide (Lupron)

Leuprolide is a 9-mer acid peptide with agonist activity against GnRH (Figure 25). It is used for the palliative treatment of prostate cancer, uterine leiomyomata, endometriosis, and central precocious puberty [130].

Leuprolide binds to GnRH receptors, triggering the downstream secretion of LH and FSH. This sustained stimulation leads to the substantial and prolonged suppression of both hormones, causing tissues and functions reliant on gonadal steroids to become quiescent [130,131,132,133].

Tap Pharmaceuticals developed Lupron, and it received FDA approval in 1995 [134]. Lupron is administered intramuscularly and may cause side effects such as itching, rashes, or hives; difficulty breathing or swallowing; pain in the arms, back, chest, neck, or jaw; slow or difficult speech; dizziness or fainting; weakness; numbness or an inability to move an arm or leg; and bone pain [130].

#### 6.1.3. Nafarelin (Synarel)

Nafarelin is a 10-mer peptide amide and GnRH agonist (Figure 26). It is used to manage endometriosis, providing pain relief and reducing endometriotic lesions [135].

Nafarelin stimulates the release of pituitary gonadotropins, LH, and FSH, causing a temporary increase in gonadal steroidogenesis. With repeated dosing, the stimulatory effect on the pituitary gland diminishes, resulting in a decreased secretion of gonadal steroids. Consequently, tissues and functions relying on gonadal steroids for maintenance become quiescent [135]. Searle developed Synarel, and it obtained FDA approval in 1998 [136]. 

Synarel is administered as a nasal solution, and it may lead to adverse effects such as shortness of breath, chest pain, urticaria, rash, and pruritus [135].

#### 6.1.4. Trelstar (Triptorelin)

Triptorelin is a 10-mer peptide amide and GnRH agonist (Figure 27). It is used for the palliative treatment of advanced prostate cancer [137].

Triptorelin initially induces a surge in the circulating levels of LH, FSH, testosterone, and estradiol. However, with continuous administration, there is a consistent decline in the secretion of LH and FSH, coupled with a notable decrease in testicular steroidogenesis. This leads to a lowering of serum testosterone concentration, reaching levels typically seen in surgically castrated men. As a result, tissues and functions reliant on these hormones for maintenance enter a quiescent state [137,138]. 

Watson Laboratories developed Triptorelin, and it received FDA approval in 2000 [139]. Triptorelin is administered intramuscularly and has been associated with side effects such as hot flushes, skeletal pain, impotence, headache, edema in legs, leg pain, erectile dysfunction, and testicular atrophy [137].

#### 6.1.5. Histrelin (Supprelin LA)

Histrelin is a 9-mer peptide amide and GnRH agonist (Figure 28). It is used for the treatment of central precocious puberty (CPP) in children [140]. 

Histrelin, as a GnRH agonist, acts as a gonadotropin inhibitor when administered continuously. This leads to an initial rise in LH and FSH, subsequently elevating gonadal steroids such as testosterone and dihydrotestosterone in males and estrone and estradiol in premenopausal females [141]. Continuous administration eventually results in the down-regulation of GnRH receptors in the pituitary gland and the desensitization of pituitary gonadotropes. This leads to decreased levels of LH and FSH [140]. Histrelin has demonstrated 100-fold higher activity than native GnRH in stimulating the release of LH hormone from monolayers of dispersed rat pituitary cells, and it exhibits a 20-fold higher potency in displacing ^125^I-GnRH from pituitary receptor sites [137].

Indevus Pharmaceuticals developed Supprelin LA, and it obtained FDA approval in 2007 [142]. Supprelin LA is administered subcutaneously and may lead to adverse effects such as implant-site reactions and the suppression of endogenous sex steroid secretion [140].

### 6.2. GnRH Antagonists

#### 6.2.1. Ganirelix (Antagon)

Ganirelix is a 10-mer peptide amide with antagonistic activity against GnRH. Ganirelix shares the same sequence as the native GnRH (Ac-pEHWSYGLRPGNH_2_), with the exception of amino acids, 1, 2, 3, 6, 8, and 10 (Figure 29) [143,144]. It is used as a fertility medicine to prevent premature LH surges or ovulation in women undergoing fertility treatment involving controlled ovarian hyperstimulation [144].

Ganirelix modulates the hypothalamic–pituitary–gonadal axis by inducing a rapid, profound, and reversible suppression of endogenous gonadotropins. During controlled ovarian stimulation, it specifically suppresses LH and FSH secretion from the pituitary gland [144]. Interestingly, unlike other GnRH agonists, ganirelix does not cause this initial increase gonadotropin levels before inhibiting premature LH surges [145]. Ganirelix does not exhibit hypo-estrogenic side effects, flare-up, or a long down-regulation period, which are commonly associated with GnRH agonists [127].

Organon developed Antagon, which received FDA approval in 1999 [146]. Antagon is administered subcutaneously and has shown various adverse effects including headache, hot flashes, pain, redness, or irritation at the injection site [144].

#### 6.2.2. Cetrorelix (Cetrotide)

Cetrorelix is a 10-mer peptide amide with antagonistic activity against GnRH [147]. Cetrorelix shares the same sequence as the native GnRH (Ac-pEHWSYGLRPGNH_2_), with the exception of amino acids, 1, 2, 3, 6, and 10 (Figure 30) [143,147]. It is used to prevent premature ovulation, which refers to the early release of eggs from the ovary [147].

Cetrorelix binds to the GnRH receptor, inhibiting the secretion of gonadotropin and thereby controlling the release of LH and FSH in a dose-dependent manner [147,148].

ASTA medica developed Cetrotide, which received FDA approval in 2000 [149]. Cetrotide is administered subcutaneously and has been associated with adverse effects such as stomach pain; bloating; nausea; vomiting; diarrhea; rapid weight gain (especially in the face and midsection); little or no urination; pain during breathing; rapid heart rate; and a sensation of breathlessness, particularly when lying down [147].

#### 6.2.3. Abarelix (Plenaxis)

Abarelix is a 10-mer linear peptide amide with potent antagonistic activity against GnRH (Figure 31). It is used in the palliative treatment of advanced prostate cancer [150].

Abarelix inhibits gonadotropin by competitively blocking GnRH receptors in the pituitary, thereby suppressing LH and FSH. This results in a reduction in the secretion of testosterone by the testicles, leading to the slowing of the growth and reduction in the size of prostate cancers [150,151]. Most importantly, abarelix does not induce the initial testosterone surge that is typical of GnRH agonists, a characteristic that could potentially exacerbate the situation [151].

Praecis Pharmaceuticals developed Plenaxis, and it received FDA approval in 2003 [152]. Plenaxis is administered intramuscularly and may result in adverse effects such as hot flashes, sleep disturbances, and breast enlargement or pain [150].

#### 6.2.4. Degarelix (Firmagon)

Degarelix is a 10-mer linear peptide amide with potent antagonistic activity against GnRH (Figure 32). It is used to treat advanced prostate cancer [153].

Degarelix inhibits GnRH receptors in the pituitary, thereby suppressing LH and FSH. This suppression leads to a reduction in the secretion of testosterone, which, in turn, slows the growth and reduces the size of prostate cancers [154]. Notably, degarelix does not induce the initial testosterone surge that is characteristic of GnRH agonists

Ferring Pharmaceuticals developed Firmagon, and it received FDA approval in 2008 [154]. Firmagon is administered subcutaneously and may lead to side reactions such as injection-site reactions, hot flashes, weight gain, and an increase in the serum levels of transaminases and γ-glutamyl transferase (GGT) [153].

## 7. Other Peptide-Based Anticancer Drugs

### 7.1. Bortezomib (Velcade)

Bortezomib is a dipeptidyl boronic acid (Figure 33). It is used for the treatment of a cancer type known as multiple myeloma [155].

Bortezomib acts as a reversible inhibitor of the chymotrypsin-like activity of the 26S proteasome. This inhibition prevents the targeted proteolysis of ubiquitinated proteins, affecting various cellular cascades and ultimately leading to cell death [155,156,157].

Millennium Pharmaceuticals developed Velcade, which obtained FDA approval in 2003 [158]. Velcade is administered intravenously and may be associated with adverse effects such as bruising and bleeding, anemia, numb or tingling hands or feet (peripheral neuropathy), diarrhea, constipation, feeling sick, and a loss of appetite [155].

### 7.2. Carfilzomib (Kyprolis)

Carfilzomib is a modified peptide epoxide consisting of four amino acids (Figure 34). It functions as a proteasome inhibitor and is employed in the treatment of multiple myeloma.

Carfilzomib irreversibly binds to the *N*-terminal of the Thr-containing active surface of the 20S proteasome, the proteolytic core particle within the 26S proteasome [159]. Its most effective target in decreasing cellular proliferation is the inhibition of the chymotrypsin-like site by carfilzomib (β5 and β5i subunits). This leads to cell cycle arrest and the apoptosis of cancerous cells. Higher doses of carfilzomib can also inhibit other catalytic active sites, including trypsin and caspase-like sites [159].

Onyx Pharmaceuticals developed Kyprolis, which was a wholly owned subsidiary of Amgen at the time. Kyprolis received FDA approval in 2012 [160]. Kyprolis is administered intravenously and may be associated with adverse effects including fatigue, anemia, nausea, thrombocytopenia, dyspnea, diarrhea, and pyrexia [159].

### 7.3. Melphalan Flufenamide (Pepaxto)

Melphalan flufenamide is an ethyl ester lipophilic peptide-inspired amide derived from melphalan (Figure 35). It is used as an alkylating drug in combination with dexamethasone for the treatment of adult patients with relapsed or refractory multiple myeloma. This applies to individuals who have undergone at least four prior lines of therapy and whose disease is refractory to at least one proteasome inhibitor, one immunomodulatory agent, and one CD38-directed mAb [161].

The notable lipophilicity of melphalan flufenamide enables its passive distribution into cells, where it undergoes metabolism to melphalan. Its antitumor activity is attributed to the crosslinking of DNA in tumor cells, specifically from the N-7 position of one guanine to the N-7 position of another. This crosslinking hinders the separation of DNA strands, thereby impeding processes such as synthesis or transcription [161,162,163,164]. Melphalan attaches alkyl groups to the N-7 position of guanine and the N-3 position of adenine, resulting in the formation of monoadducts. DNA fragmentation occurs when repair enzymes attempt to correct this error [163,164]. Additionally, melphalan can induce further mutations [163].

Oncopeptides AB developed Pepaxto, which received FDA approval in 2021 [165]. However, due to an increased rate of death being observed in phase 3 clinical trials, melphalan flufenamide was subsequently withdrawn (discontinued) later in the same year [166]. Pepaxto is administered intravenously and has exhibited several side effects, with those occurring in more than 20% of cases including fatigue, nausea, diarrhea, pyrexia, and respiratory tract infection. The side effects seen in at least 50% of cases include decreased levels of leukocytes, platelets, lymphocytes, neutrophils, and hemoglobin and an increase in creatinine [161].

## 8. Peptide Drug Conjugates (PDCs)

PDCs are categorized as both cell-penetrating peptides (CPPs) and cell-targeting peptides (CTPs) [167]. PDCs have a small molecular weight, typically ranging from approximately 2000 to 20,000 Da. This size enables PDCs to effectively penetrate the tumor stroma and enter the tumor cells [168]. Hence, PDCs are increasingly acknowledged as a novel modality for targeted drug delivery, offering enhanced efficacy and diminished side effects in cancer treatment when compared to ADCs [169]. Furthermore, in contrast to ADCs, which have a restricted range of cytotoxic payloads such as MMAE, MMAF, and mertansine (DM-1), PDCs offer the flexibility to be coupled with various molecules exhibiting anticancer activity [170]. Similar to peptides, PDCs face challenges related to low stability and rapid clearance [167]. Nonetheless, addressing these issues is possible through the structural modification of peptides and the exploration of innovative structural motifs that enhance both stability and circulation time [171]. Mándity and coworkers have created a database of PDCs (ConjuPepBD) that encompasses over 1600 conjugates documented in approximately 230 publications [172]. Lutathera and Pepaxto stand as the sole FDA-approved peptide drug conjugates (PDCs) [173]. However, with the discontinuation of Pepaxto, only one approved PDC remains [166].

## 9. Conclusions

Peptides are an important class of compounds that are increasingly gaining FDA approvals, contributing significantly to addressing unmet medical challenges. Due to their substantial binding footprint with therapeutic targets, peptides are considered superior to their small-molecule counterparts. Despite facing stability challenges against enzymatic degradation, ongoing harmonization efforts across interdisciplinary fields have played a vital role in enhancing the pharmacokinetic and pharmacodynamic profiles of peptides [171]. 

Several diagnostic peptides are continually gaining FDA approval, playing a pivotal role in the early detection of cancer, facilitating more effective treatment. Moreover, the theragnostic class with the dual capability holds paramount significance for the prompt treatment of cancer. ADCs provide the potential to deliver cytotoxic agents directly to the tumor cells (on-site delivery), minimizing the risk of off-target adverse effects on healthy cells. 

Peptides offer higher solubility, a greater safety profile, and greater selectivity, while small molecules provide high oral availability, metabolic stability, and high membrane permeability [174]. In such situations, PDCs provide a valuable alternative wherein peptides are employed as carriers to deliver small molecule-based cytotoxic agents [174].

Oral peptide delivery approaches hold potential advantages over parenteral, inhalation, or intranasal delivery strategies. However, solid-state dosage forms for oral peptide delivery encounter significant challenges in their manufacture, stability, and delivery which currently constrain their effective deployment. Out of the 29 peptides reviewed in this article, only 1 can be administered orally, accounting for a mere 3.4% of the total approvals. Various tactics are considered to enhance the stability of peptides, but no novel approaches are in place [171,175].

A UK patent has recently been filed by our research group, showcasing the successful synthesis of first-in-class peptide analogs [176]. These peptide analogs are envisaged to provide unprecedented stability, presenting a potential opportunity for them to be orally administered. Their novel multiple-chained structure is extensively constrained, significantly enhancing overall resistance against enzymatic degradation. Additionally, the innovative design of this class enables the independent tuning of the physicochemical properties of each chain, including log*D* and *p*Ka. These two factors are critical for oral bioavailability but challenging to manage in common peptide structures.

Given the ongoing advancements in the field of peptides, I anticipate that peptides will emerge as the preferred alternatives across various drug classes, particularly as pioneering agents in anticancer treatments. The acknowledged capability of peptides to employ dual mechanisms in interacting with therapeutic targets is noteworthy. The FDA is increasingly approving various first-in-class peptides, demonstrating their potential to effectively treat rare genetic diseases like Rett syndrome [13]. The FDA’s recognition of these properties is evident through the growing number of approvals for such molecules [177]. Notably, an echinocandin peptide analog received FDA approval in 2023 despite its initial approval dating back to 1955, highlighting the enduring effectiveness of peptides in combating diseases [13]. Intriguingly, in the pharmaceutical landscape, peptides do not merely outpace other drug classes; they can also transport and deposit these classes precisely at the tumor site or even within the tumor itself. Consequently, leveraging the advantages of this conjugate while addressing its drawbacks has the potential to enhance the overall treatment process. ADC and PDC provide a notable illustration of how peptides exhibit such functional capabilities.

It is hoped that continued efforts from all disciplines will result in the development of better ways to fight cancer and promote a healthier and improved life for people around the world.

## Figures and Tables

**Figure 1 cancers-16-01032-f001:**
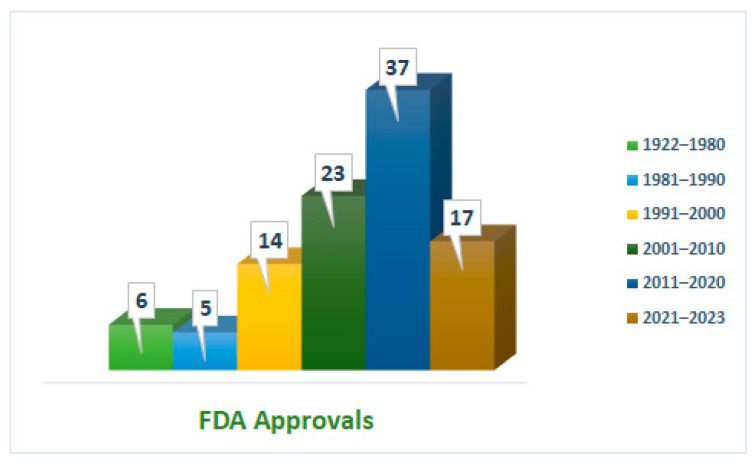
FDA-approved peptides from 1922 to 2023. Each bar represents a period of 10 years; however, due to the limited number of approvals, for the period between 1922 and 1980, one bar was assigned.

**Figure 2 cancers-16-01032-f002:**
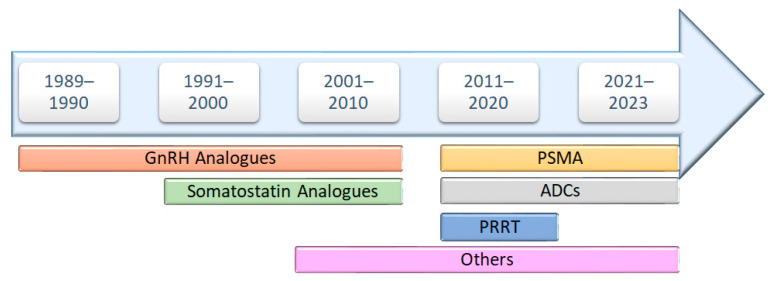
Development timelines of peptide-based anticancer drugs.

**Figure 3 cancers-16-01032-f003:**
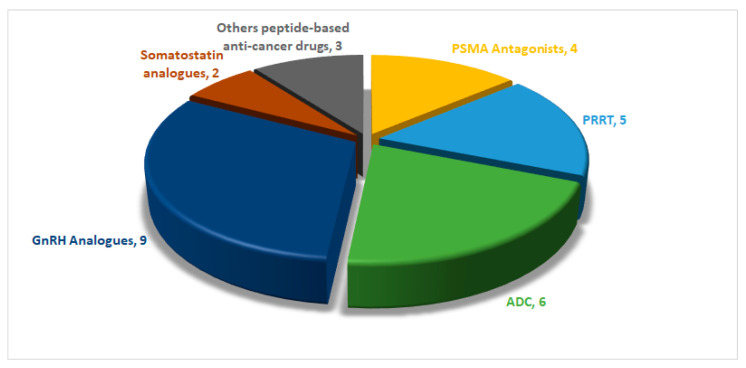
FDA-approved anticancer peptides (1989–2023). ADC, antibody drug conjugate; GnRH, gonadotropin-releasing hormone; PRRT, peptide receptor radionuclide therapy; PSMA, prostate-specific membrane antigen.

**Figure 4 cancers-16-01032-f004:**
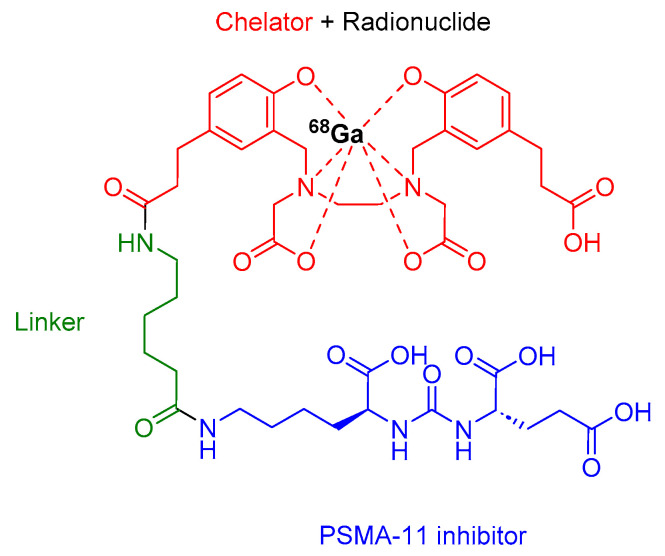
Chemical structure of ^68^Ga-PSMA-11. Blue, PSMA inhibitor; green, linker; red, chelator; black, radionuclide.

**Figure 5 cancers-16-01032-f005:**
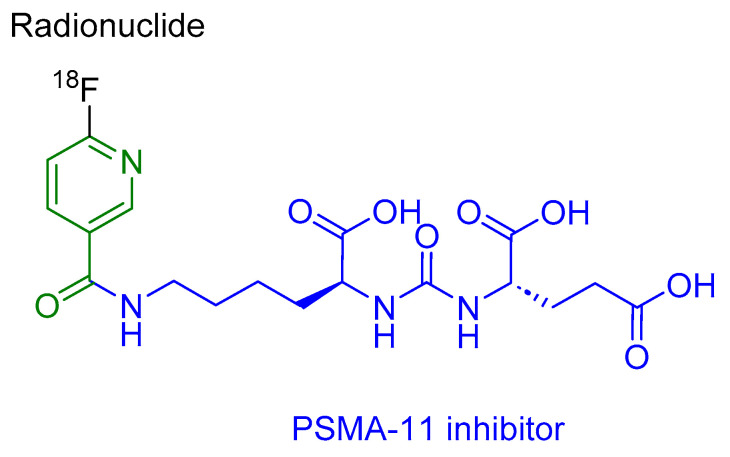
Chemical structure of piflufolastat F 18. Blue, PSMA inhibitor; green, pyridine-3-carbonyl; black, radionuclide.

**Figure 6 cancers-16-01032-f006:**
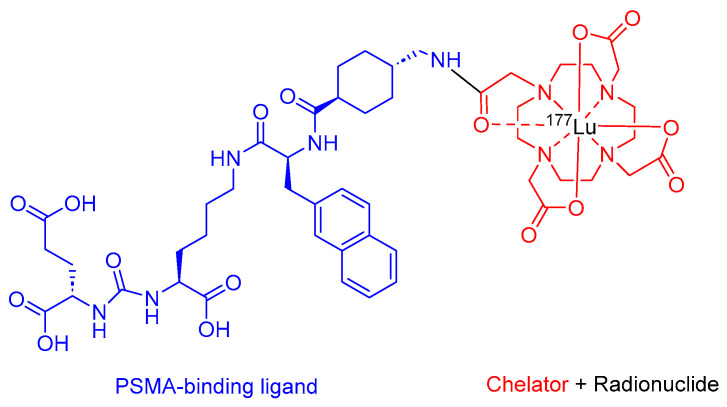
Chemical structure of Pluvicto. Blue, PSMA inhibitor; red, chelator; black, radionuclide.

**Figure 7 cancers-16-01032-f007:**
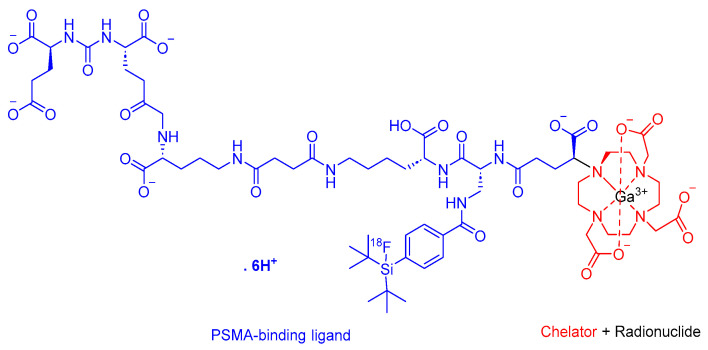
Chemical structure of flotufolastat F 18. Blue, PSMA inhibitor; red, chelator; black, radionuclide.

**Figure 8 cancers-16-01032-f008:**
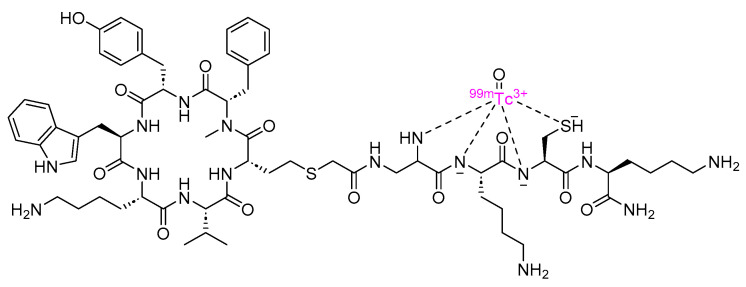
Chemical structure of depreotide. Pink, radionuclide.

**Figure 9 cancers-16-01032-f009:**
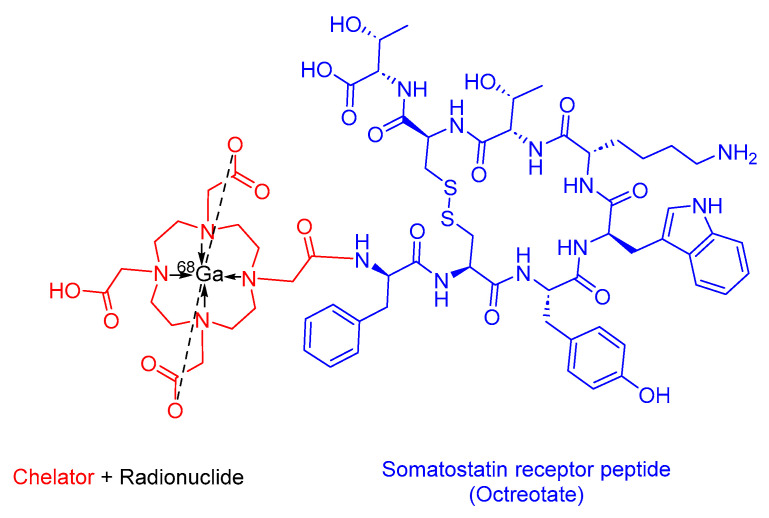
Chemical structure of ^68^Ga-DOTATATE. Blue, octreotate; red, chelator; black, radionuclide.

**Figure 10 cancers-16-01032-f010:**
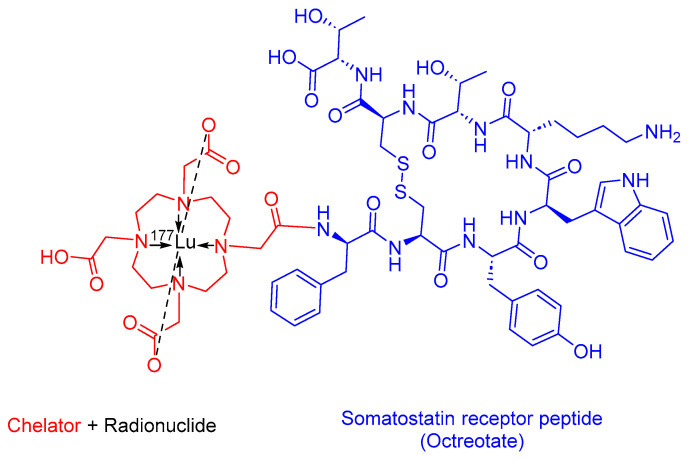
Chemical structure of [^177^Lutetium]Lu-DOTA-TATE. Blue, octreotate; red, chelator; black, radionuclide.

**Figure 11 cancers-16-01032-f011:**
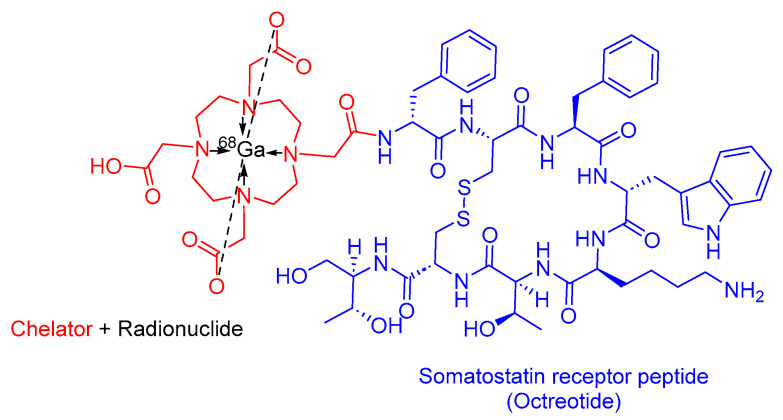
Chemical structure of ^68^Ga-DOTATOC. Blue, octreotide; red, chelator; black, radionuclide.

**Figure 12 cancers-16-01032-f012:**
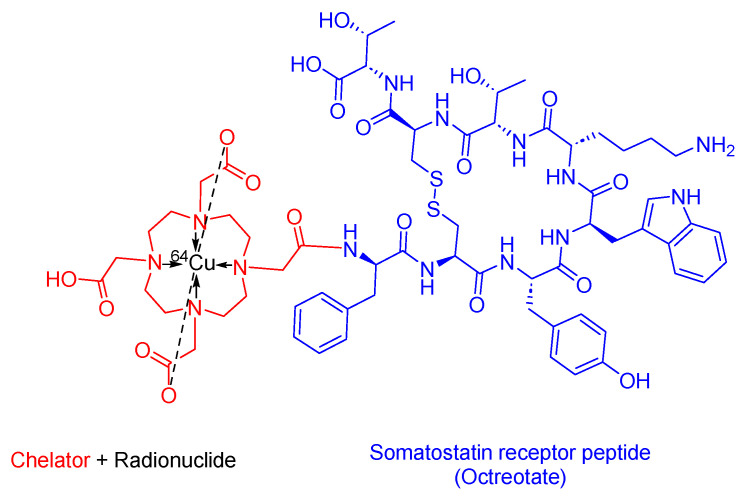
Chemical structure of ^64^Cu-DOTATATE. Blue, octreotate; red, chelator; black, radionuclide.

**Figure 13 cancers-16-01032-f013:**
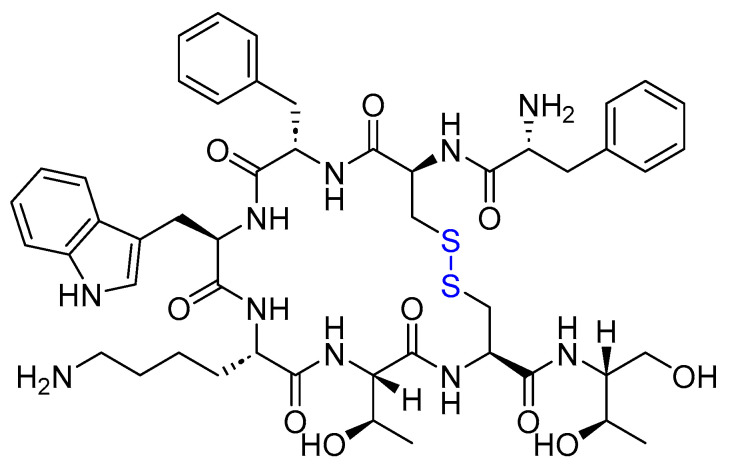
Chemical structure of octreotide. Blue: disulfide bridge.

**Figure 14 cancers-16-01032-f014:**
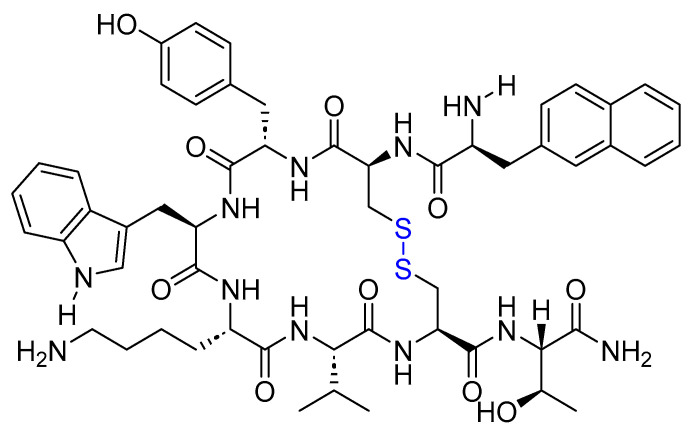
Chemical structure of Lanreotide. Blue: disulfide bridge.

**Figure 15 cancers-16-01032-f015:**
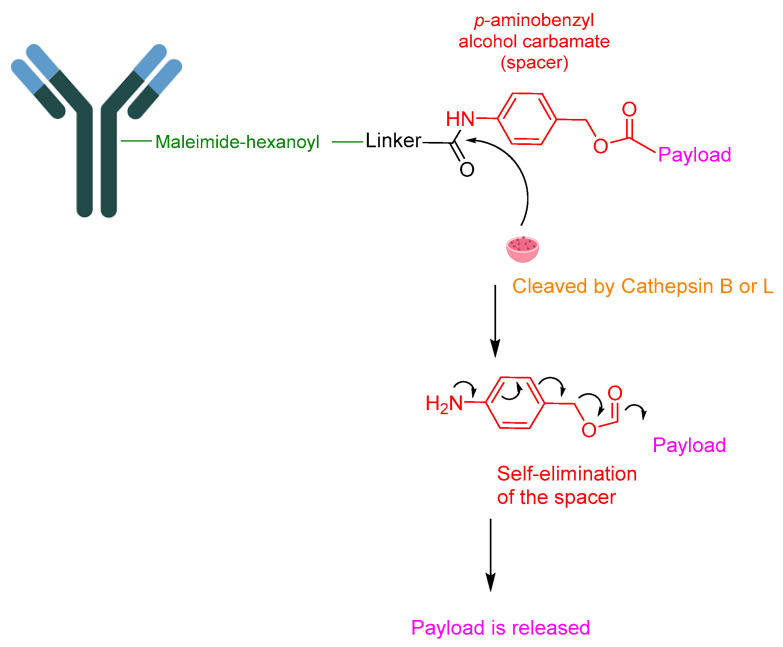
Payload release mechanism in ADCs. Cathepsin B for Val-Cit, cathepsin B and cathepsin L for GGFG linker. Green, maleimide (caproyl) hexanoyl; black, linker; red, spacer; pink, payload; Orange, cathepsin B or cathepsin L.

**Figure 16 cancers-16-01032-f016:**
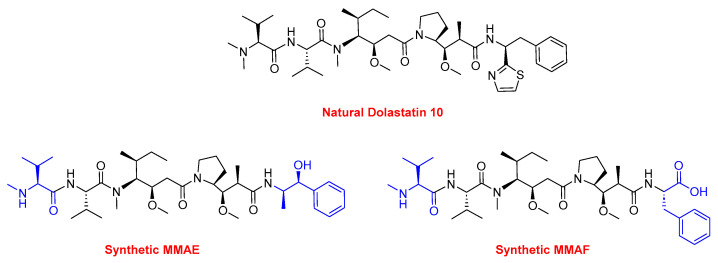
The chemical structure of the natural dolastatin 10, MMAE, and MMAF. Blue: differences from the natural dolastatin 10.

**Figure 17 cancers-16-01032-f017:**
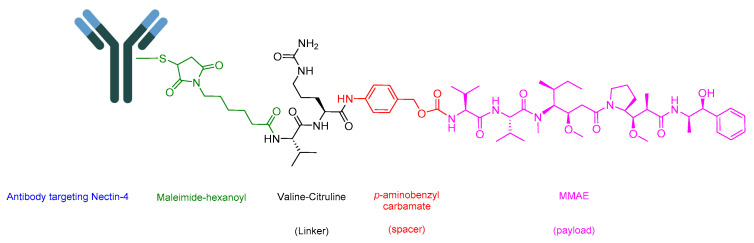
Chemical structure of enfortumab vedotin-ejfv. Green, maleimide (caproyl) hexanoyl; black, linker; red, spacer; pink, payload.

**Figure 18 cancers-16-01032-f018:**
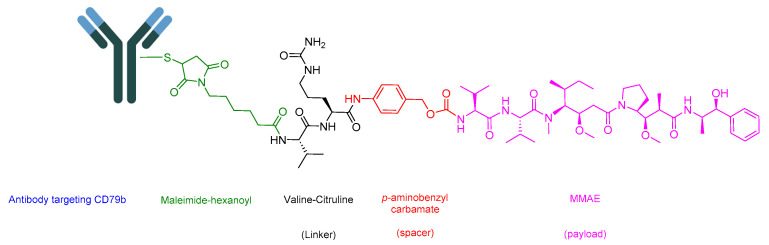
Chemical structure of polatuzumab vedotin-piiq. Green, maleimide (caproyl) hexanoyl; black, linker; red, spacer; pink, payload.

**Figure 19 cancers-16-01032-f019:**
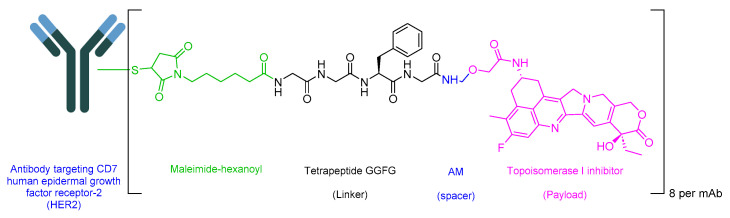
Chemical structure of fam-trastuzumab deruxtecan-nxki. Green, maleimide (caproyl) hexanoyl; black, linker; blue, spacer; pink, payload.

**Figure 20 cancers-16-01032-f020:**
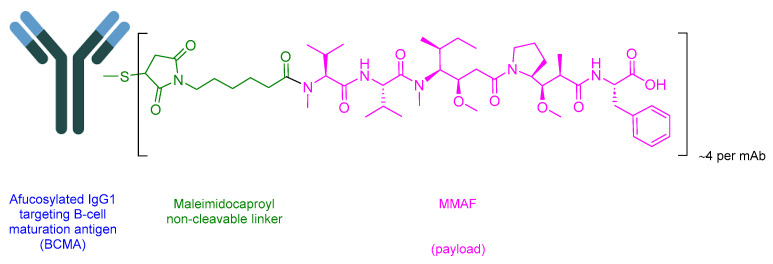
Chemical structure of belantamab mafodotin-blmf. Green, maleimide (caproyl) hexanoyl; pink, payload.

**Figure 21 cancers-16-01032-f021:**
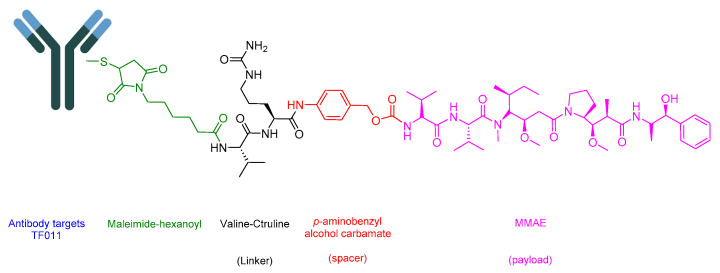
Chemical structure of tisotumab vedotin-tftv. Green, maleimide (caproyl) hexanoyl; black, linker; red, spacer; pink, payload.

**Figure 22 cancers-16-01032-f022:**
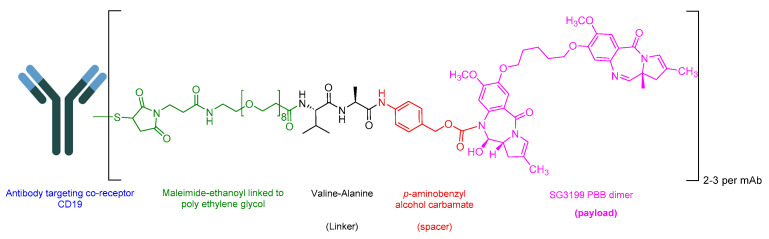
Chemical structure of loncastuximab tesirine-lpyl. Green, maleimide ethanoyl linked to polyethylene glycol; black, linker; red, spacer; pink, payload.

**Figure 23 cancers-16-01032-f023:**
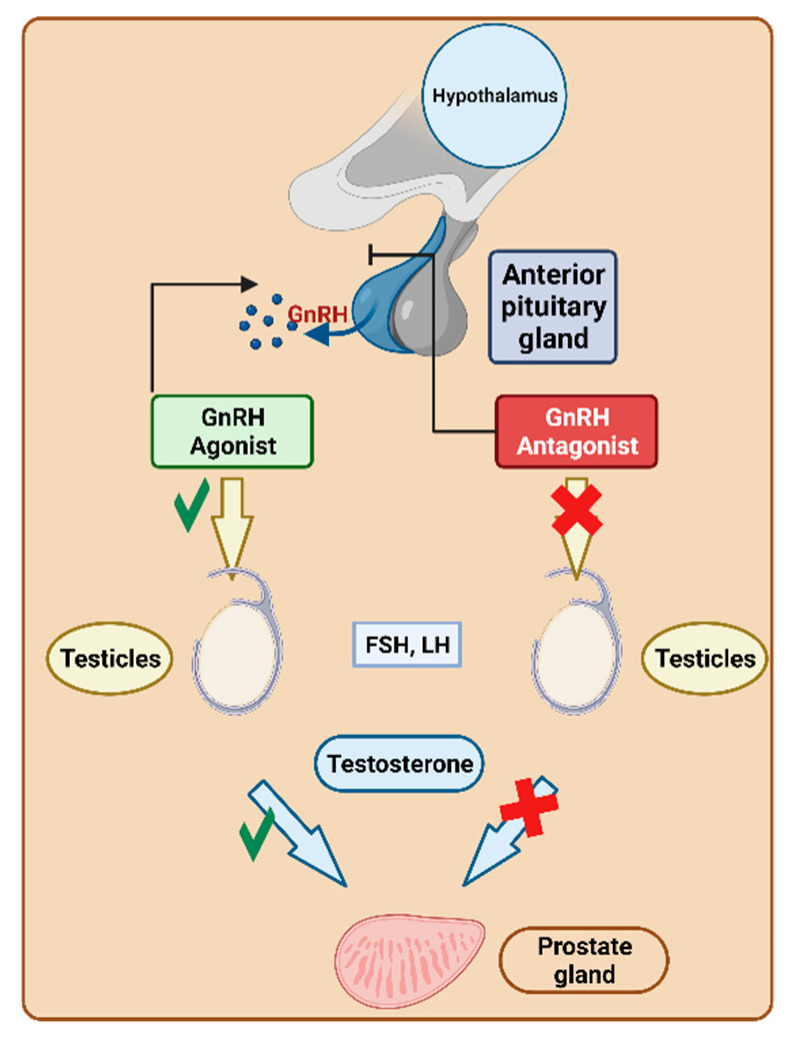
Gonadotropin-releasing hormone (GnRH) mechanism of action. LH, luteinizing hormone; FSH, follicle-stimulating hormone. Adapted from [124].

**Figure 24 cancers-16-01032-f024:**
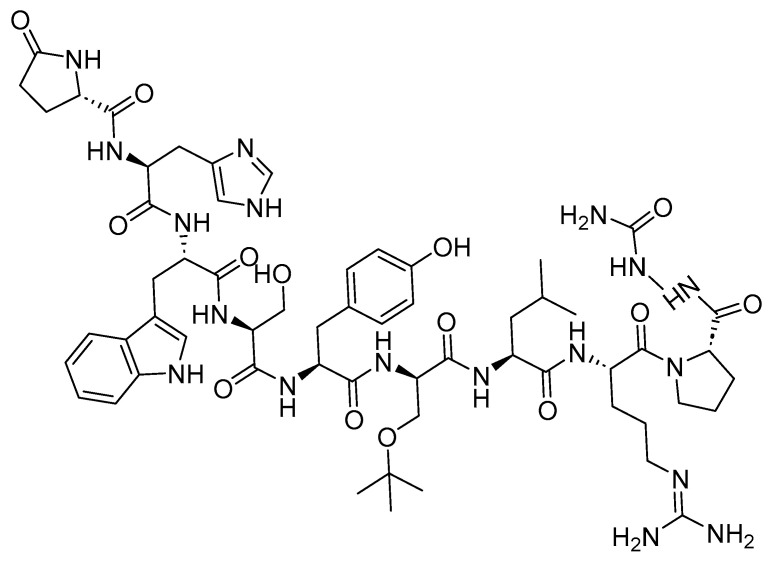
Chemical structure of goserelin.

**Figure 25 cancers-16-01032-f025:**
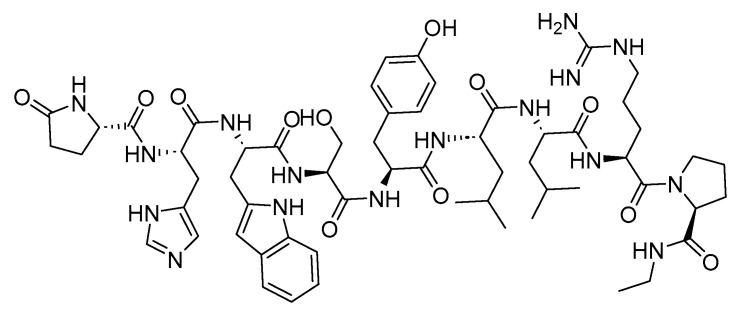
Chemical structure of leuprolide.

**Figure 26 cancers-16-01032-f026:**
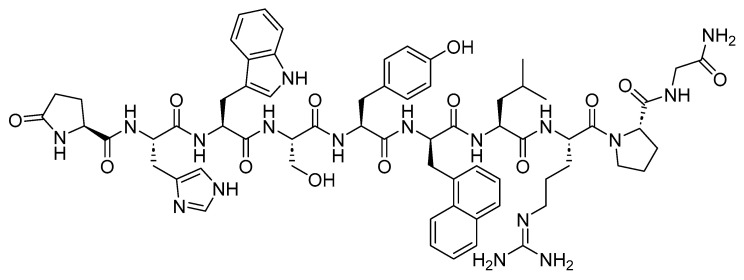
Chemical structure of nafarelin.

**Figure 27 cancers-16-01032-f027:**
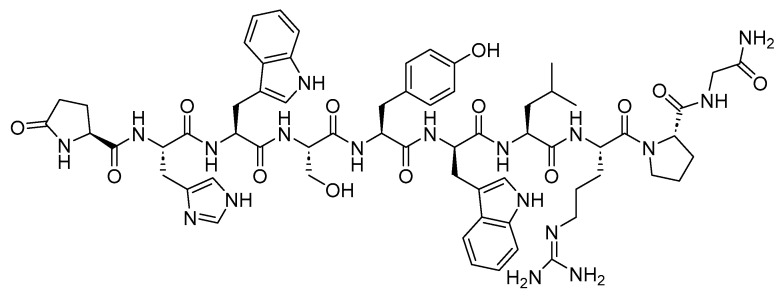
Chemical structure of triptorelin.

**Figure 28 cancers-16-01032-f028:**
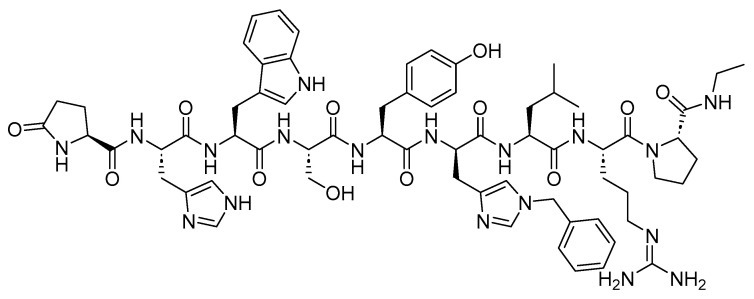
Chemical structure of histrelin.

**Figure 29 cancers-16-01032-f029:**
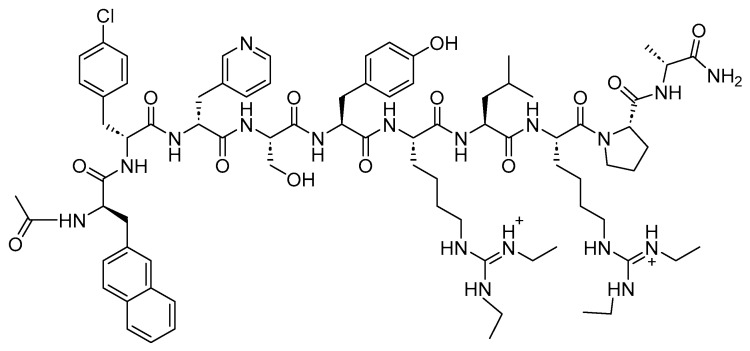
Chemical structure of ganirelix.

**Figure 30 cancers-16-01032-f030:**
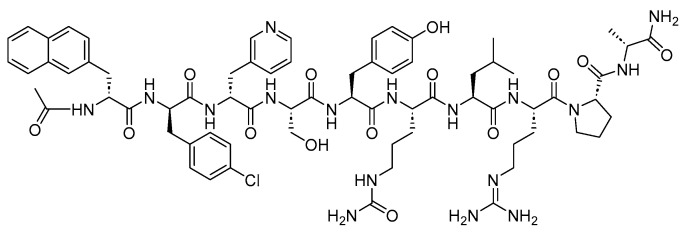
Chemical structure of cetrorelix.

**Figure 31 cancers-16-01032-f031:**
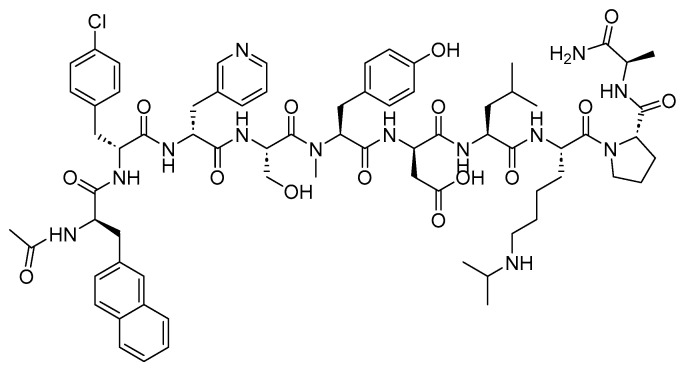
Chemical structure of abarelix.

**Figure 32 cancers-16-01032-f032:**
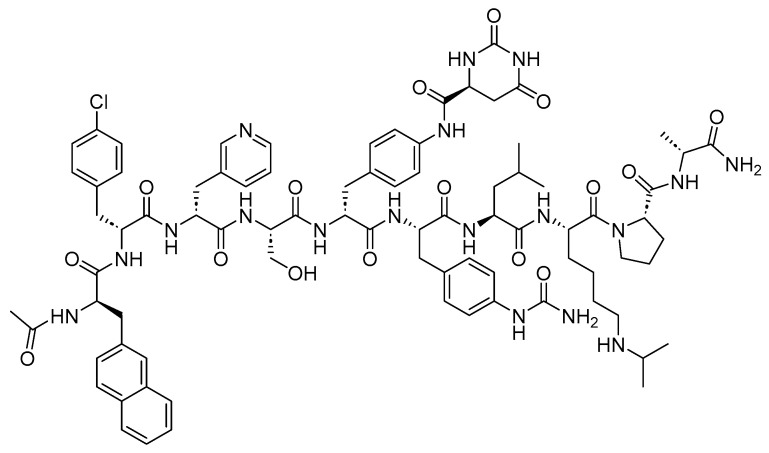
Chemical structure of degarelix.

**Figure 33 cancers-16-01032-f033:**
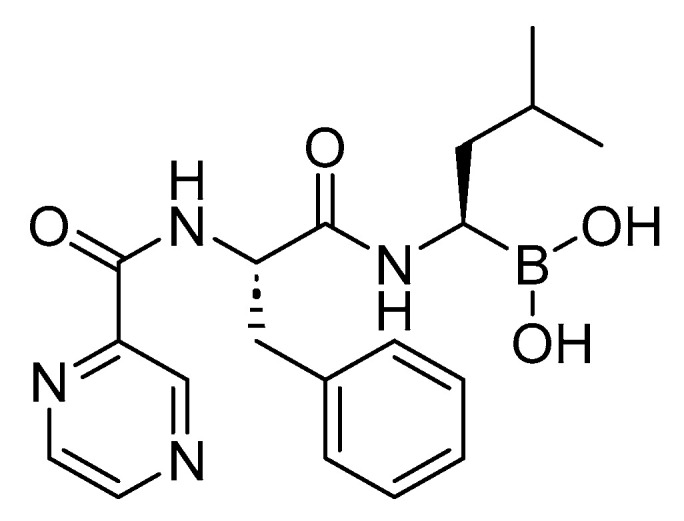
Chemical structure of bortezomib.

**Figure 34 cancers-16-01032-f034:**
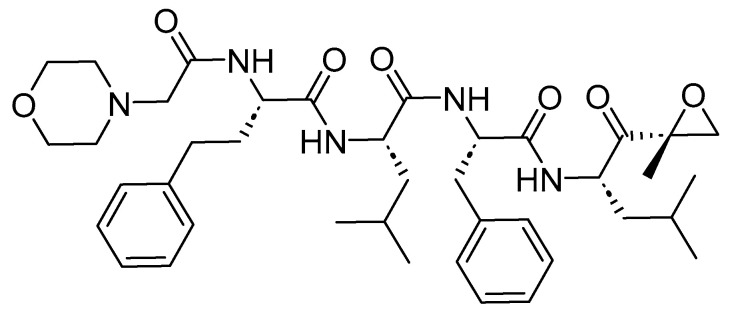
Chemical structure of carfilzomib.

**Figure 35 cancers-16-01032-f035:**
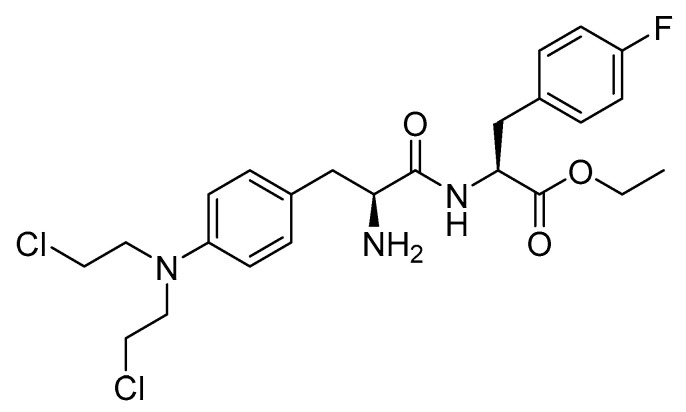
Chemical structure of melphalan flufenamide.

**Table 1 cancers-16-01032-t001:** FDA-approved prostate-specific membrane antigen (PSMA).

Peptide (Trade Name)	Indication	Therapeutic Target	Route	FDA Approval Year
^68^Ga-PSMA-11 (^68^Ga gozetotide)	A diagnostic employed for detecting PET PSMA-positive lesions in males diagnosed with prostate cancer	PSMA	IV	2020
Piflufolastat F 18 (Pylarify)	A diagnostic employed for detecting PET PSMA-positive lesions in males diagnosed with prostate cancer	PSMA	IV	2021
Lutetium ^177^Lu Vipivotide Tetraxetan (Pluvicto)	Treatment of PSMA-positive metastatic castration-resistant prostate cancer (mCRPC) in adult patients	PSMA and the neighboring cells	IV	2022
Flotufolastat F 18 (Posluma)	A diagnostic employed for detecting PET PSMA-positive lesions in males diagnosed with prostate cancer	PSMA	IV	2023

IV, intravenous; PET, positron emission tomography; PSMA, prostate-specific membrane antigen.

**Table 2 cancers-16-01032-t002:** FDA-approved peptide receptor radionuclide therapies (PRRTs).

Peptide (Trade Name)	Indication	Therapeutic Target	Route	FDA Approval Year
Depreotide (Neotect)	Scintigraphic imaging	Somatostatin receptor	IV	1999
^68^Ga-DOTATATE (Netspot)	Scintigraphic imaging	Somatostatin receptor	IV	2016
^177^Lu-DOTATATE (Lutathera)	To treat somatostatin receptor-positive GEP-NETs, including foregut, midgut, and hindgut NETs.	Somatostatin receptor	IV	2018
^68^Ga-DOTATOC	Scintigraphic imaging	Somatostatin receptor	IV	2019
^64^Cu-DOTATATE (Detectnet)	Scintigraphic imaging	Somatostatin receptor	IV	2020

GEP-NETs, gastroenteropancreatic neuroendocrine tumors; IV, intravenous.

**Table 3 cancers-16-01032-t003:** FDA-approved antibody drug conjugates (ADCs).

Peptide (Trade Name)	Indication	Therapeutic Target	Route	FDA Approval Year
Enfortumab Vedotin-Ejfv (Padcev)	Urothelial cancer (a cancer of the bladder and urinary tract).	Nectin-4 receptor	IV	2019
Polatuzumab vedotin-piiq (Polivy)	1. DLBCL patients whose cancer has returned or has stopped responding to other treatments and who cannot have a bone-marrow transplantation. 2. Relapsed or refractory DLBCL, NOS, after undergoing at least two prior therapies interventions.	CD79b	IV	2019
Fam-trastuzumab deruxtecan-nxki (Enhertu)	Unresectable or metastatic HER2-positive breast cancer and concomitantly has unresectable or metastatic NSCLC.	Human epidermal growth factor receptor-2 (HER2)	IV	2019
Belantamab Mafodotin-Blmf (Blenrep)	Relapsed or refractory multiple myeloma.	B-cell maturation antigen (BCMA)	IV	2020
Tisotumab Vedotin-Tftv (TIVDAK)	Recurrent or metastatic cervical cancer with disease progression following chemotherapy.	Tissue factor TF011	IV	2021
Loncastuximab Tesirine-Lpyl (Zynlonta)	Relapsed or refractory large B-cell lymphoma after two or more lines of systemic therapy, including DLBCL.	CD19	IV	2021

DLBCL, diffuse large B-cell lymphoma; IV, intravenous; NOS, not otherwise specified; NSCLC, non-small-cell lung cancer.

**Table 4 cancers-16-01032-t004:** FDA-approved gonadotropin-releasing hormone (GnRH) peptides.

Peptide (Trade Name)	Indication	Therapeutic Target	Route	FDA Approval Year
**Agonists**				
Goserelin (Zoladex)	Managing carcinoma of the prostate, addressing endometriosis, and providing palliative treatment for advanced breast cancer	GnRH	SC	1989
Leuprolide (Lupron)	Palliative treatment for prostate cancer, uterine leiomyomata, endometriosis, and central precocious puberty	GnRH	IM	1995
Nafarelin (Synarel)	To address endometriosis, including the alleviation of pain and the reduction of endometriotic lesions	GnRH	Nasal solution	1998
Trelstar (triptorelin)	Palliative treatment of advanced prostate cancer	GnRH	IM	2000
Histrelin (Supprelin LA)	Treatment of central precocious puberty (CPP) in children	GnRH	SC	2007
		GnRH		
**Antagonists**				
Ganirelix (Antagon)	A fertility medication designed to prevent premature LH surges or ovulation in women undergoing fertility treatment with controlled ovarian hyperstimulation	GnRH	SC	1999
Cetrorelix (Cetrotide)	To inhibit premature ovulation	GnRH	SC	2000
Abarelix (Plenaxis)	Palliative treatment of advanced prostate cancer	GnRH	IM	2003
Degarelix (Firmagon)	Advanced prostate cancer	GnRH	SC	2008

GnRH, gonadotropin-releasing hormone; IM, intramuscular; LH, luteinizing hormone; SC, subcutaneous.

## Data Availability

Not applicable.

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
