# Peer review of "Peptide Therapeutics: Unveiling the Potential against Cancer—A Journey through 1989"

_cancers, 2024, doi:10.3390/cancers16051032_

Round 1

Reviewer 1 Report

Comments and Suggestions for Authors

Report attached

Comments on the Quality of English Language

Recheck for some typo errors.

Author Response

I attached my response. Thanks!

Othman

Reviewer 2 Report

Comments and Suggestions for Authors

This review provides a detailed introduction to the application of peptides in cancer treatment. In this review, prostate-specific membrane antigen (PSMA) peptide antagonists, peptide receptor radionuclide therapy (PRRT), antibody-drug conjugate (ADC), gonadotro-pin-releasing hormone (GnRH) analogues, somatostatin analogues, and other peptide-based anti-cancer drugs are analysed in terms of their chemical structures and properties, therapeutic target and mechanism of action, development journey, administration route, and side effects. But there are still some minor errors that need to be corrected:

1.      Some descriptions of the relationship between peptides and cancer treatment should be added to the Simple Summary to make the content of the article clearer.

2.      An extra parenthesis is added after “chemical synthesis”. (Page 2, paragraph 4, line 1)

3.      Chemical synthesis should not be a source of peptides, but rather a means of synthesizing peptides. (Page 2, paragraph 4, line 1)

4.      "Can be derived by" can be changed to "can be derived from" to match the context. (Page 2, paragraph 4, line 1)

5.      The f in flotufolastat should be capitalized. (Page 5, paragraph 3, line 1)

6.      What does “10-m34 peptide amide” mean? (Page 16, paragraph 3, line 1)

7.      Missing a period at the end of the sentence. (Page 20, paragraph 1, line 1)

8.      Images and legends should be centered in the row.

9.      There is an extra period before the reference. (Page 23, paragraph 2, line 3; Page 23, paragraph 3, line 7)

10.   The “p” and “D” in “logD and pKa” need to be in italics. (Page 23, paragraph 2, line 3)

Comments on the Quality of English Language

none

Author Response

I attached my response. Thanks!

Othman

Reviewer 3 Report

Comments and Suggestions for Authors

The manuscript entitled "Peptide Warriors: Unveiling the Potential Against Cancer - A Journey Through 1989" is written as per the style.

However, a few minor corrections and edits are required, which are mentioned in the annotated PDF file. The plagiarism is detected, and a few paragraphs require modification. The language correction is required. The capitalization of a few abbreviations is required. The abbreviations should be listed in the table.

Please refer PDF files

Comments on the Quality of English Language

Minor English language correction is required.

Author Response

I attached my response. Thanks!

The required modifications are highlighted in Turquoise in the manuscript

Othman

Reviewer 4 Report

Comments and Suggestions for Authors

In the last decades many efforts have been devoted in creating new therapies that are at the same time more selective and less harmful for the patients. Despite this, the methods today available such as surgery, chemotherapy and radiation, and combination of those method, have a relatively low success rate as well as they present a risk of recurrence. A way to improve chemotherapy is to develop new strategies such as drug modifications and development of new drug-carrier systems in order to enhance the tumor targeting properties of anticancer drugs. Peptides are highly selective and efficacious, relatively safe and highly tolerated. Their metabolism (biodegradation) is predictable, their synthesis on solid phase is relatively easy and allows modifications of such features as hydrophobicity, affinity, charge, solubility and stability, as well as generates lower costs than production of antibodies. On the other hand, for a long time bioactive peptides were excluded from the group of potent drug candidates due to their some serious intrinsic weakness - poor chemical and physical stability and short half-life in blood plasma. To overcome these disadvantages, various strategies were developed and successfully applied showing the potency of such biopharmaceuticals as effective and highly specific therapeutics of asthma, cancer, neuropathic pain, HIV, heart disease, stroke and diabetes. Like in case of other biologically active compounds, nature offers the most abundant source of leads for the discovery of peptide-based drugs. Being a product of evolutionary selection, endogenous peptides use a variety of mechanisms responsible for their biological activity, including electrostatic attraction to negatively charged microbial membranes or some cancer cells, specific receptor-mediated targeting, interfering with intracellular biochemical reactions (DNA/RNA synthesis, enzyme inhibition.

In this work, the author present usage and functions of peptides in selected FDA-approved peptide-based drugs with anti-cancer properties.

In general, this work is interesting but below are some points that authors should/could take into consideration to advance the current version of their work:

1.      I found some stylistic errors, like double spaces, but also: a.       on the second page, in the 10 line from the bottom, please remove the bracket after “synthesis” b.      on the 5 page, in the 10 line from the bottom it should be capital letter in word “flotufolastat F” c.       on the 21 page, in the line 2 from the bottom it should be capital letter in word “it”

2.      I would suggest to improve the figures captions namely by providing more detail about the images description.

3.      It would increase the value of the manuscript if the author can provide their critical view of the field instead of just describing what is being done in the field.

4.      I would suggest to add explanation and difference between ADC ( antibody drug conjugate) and PDC (peptide-drug conjugate).  It is worth to mantion that Mándity and coworkers created a freely available, fully annotated and manually curated database of peptide drug conjugates- ConjuPepBD. It covers more than 1600 conjugates from 230 publications. The database is accessible at: https://conjupepdb.ttk.hu/. The advanced linker technology utilized for ADCs may be helpful in the design and development of other drug conjugates. Compared with antibodies (mAb), smaller peptides may offer better penetration through solid tumors. Moreover, owing to the simplicity of peptides and the ease of possible reactions during PDC synthesis, it is envisioned that further technologies that have not been explored in ADCs may give rise to novel linkers and structures with the potential for enhanced efficacy, longer circulation time, and lower toxicity. Given the accumulation of technological improvements such as site-specific conjugation approaches, the generation of highly efficacious and less toxic drug conjugates is opening to unprecedented possibilities

Author Response

I attached my response. Thanks!

Othman

Round 2

Reviewer 4 Report

Comments and Suggestions for Authors

Thank you for all modification. I accpet all of this changes.